# Methods of Anterior Torque Control during Retraction: A Systematic Review

**DOI:** 10.3390/diagnostics12071611

**Published:** 2022-07-01

**Authors:** Anna Ewa Kuc, Jacek Kotuła, Marek Nahajowski, Maciej Warnecki, Joanna Lis, Ellie Amm, Beata Kawala, Michał Sarul

**Affiliations:** 1Dental Star Specjalistyczne Centrum Stomatologii Estetycznej, 15-215 Bialystok, Poland; dental.star@wp.pl; 2Department of Dentofacial Orthopedics and Orthodontics, Wroclaw Medical University, Krakowska 26, 50-425 Wroclaw, Poland; joanna.lis@umw.edu.pl (J.L.); beata.kawala@umw.edu.pl (B.K.); 3Department of Integrated Dentistry, Wroclaw Medical University, Krakowska 26, 50-425 Wroclaw, Poland; marek_nahajowski@wp.pl (M.N.); michal.sarul@umw.edu.pl (M.S.); 4Private Practice, Praxi-Dent, Niemodlinskiej 63, 45-864 Opole, Poland; maciej.warnecki@gmail.com; 5Department of Orthodontics and Dentofacial Orthopedics, Boston University, Boston, MA 02118, USA; elieamm@hotmail.com

**Keywords:** tooth retraction, torque control, orthodontics

## Abstract

Background: There are various methods of controlling the inclination of the incisors during retraction, but there is no evidence as to the advantages of some methods over others. The purpose of this systematic review and meta-analysis was to determine the effectiveness of the methods used to control torque during anterior teeth retraction. Methods: In accordance with the PRISMA guidelines, the main research question was defined in the PICO format [P: patients with complete permanent dentition; I: the maxillary incisor torque after (I) and before I retraction with straight-wire appliance and different modes of torque control; O: statistically significant differences in torque values of the upper incisors after orthodontic treatment]. The MEDLINE, EMBASE, and Cochrane Central Register of Controlled Trials databases were searched for keywords combining: retraction orthodontics, torque control orthodontics, torque orthodontics, inclination orthodontics, torque control retraction. The articles were subjected to risk of bias and quality analyses with the ROBINS-I protocol and the modified Newcastle–Ottawa QAS, respectively. Meta-analyses were performed with both fixed- and random-effects models. Results: 13 articles were selected in which total number of 580 subjects took part. In all studies, incisors were retroclined during retraction by 2.46° (mean difference), which was statistically significant. Considering the articles separately, the differences in torque between the study group and the control group were statistically significant in six articles. The Q statistic was 36.25 with *p* = 0.0003 and I^2^ = 66.9%, which indicated a high level of study heterogeneity. Conclusion: Both properly performed corticotomy and en-masse retraction using orthodontic microimplants seem to be the most effective and scientifically validated methods of torque control. Further high-quality research is needed to perform better quality analyses and draw more reliable conclusions.

## 1. Introduction

Torque control is a key element in the extraction of the first premolars during orthodontic treatment [1,2,3,4,5,6,7,8,9,10,11,12,13,14,15,16]. In such cases, it is necessary to thoroughly diagnose and plan the appropriate control of the torque. The authors tried to show which of these methods gives the greatest effectiveness [6,7,8,9,10,11,12,13,14,15,16,17,18,19].

During orthodontic treatment, torque control of the anterior teeth roots is relevant. It ensures the stability of the proper interincisal angle that, in turn, is responsible for the proper support of soft tissues [20,21,22], providing a harmonious facial profile [6,7]. The labio-palatal inclination of the long axis of the incisors is also relevant to maintaining a healthy periodontium, which minimises the risk of recurrence after treatment, recession, fenestration or dehiscence in the anterior part of the dental arch [6].

In addition to attempts to control the inclination of the incisors during their retraction directly related to the interaction of the orthodontic wire with the surface of the breech gap, a modification of the direction of the force vector can also be used. In particular, the use of orthodontic mini-implants—one of the greatest achievements of orthodontics of the last 20 years—should be taken into account here. This method can affect not only better anchor control but also the way the incisors move during their retraction [22].

There is a need for the mechanical control of the incisor root position in the treatment of moderate to severe crowding, bimaxillary protrusion [12], and open bite or Class II malocclusion. The extraction of premolars is often necessary to distalize canines into good position. Spaces occurring mesially to distalised canines make the incisors particularly susceptible to uncontrolled/excessive inclination. To avoid this effect, such different methods of incisor torque control were suggested as brackets with increased built-in torque [7], arch torsion [12], placement of temporary intraoral skeletal anchorage devices (TISAD) [8,9,10,11,14,15,16,17] that enable group distal movement of the “social six” (en-masse retraction) [8,12,13], and a mini-implant inserted between the maxillary incisors [9,14,15]. Regardless of various procedures that support proper root position during space closure, the evaluation of their effectiveness is based almost exclusively on individual clinicians’ experience.

Therefore, this systematic review aims to objectively determine the effectiveness of different methods for root torque control of the maxillary incisors during their orthodontic retraction, and thus to identify which of the suggested procedures deserve the highest recommendations in clinical practice.

## 2. Methods

The systematic review was registered in the PROSPERO database under identification number CRD42021215408.

The study was conducted according to PRISMA (Preferred Reporting Items for Systematic Reviews and Meta-Analyses) guidelines [3,4]. The research questions were defined in PICO format:

Population (P): The patients undergoing the study had complete permanent dentition.

Intervention (I): The orthodontic extraction treatment with braces using a straight arch technique and an additional method for incisor root torque control was used.

Comparison (C): Evaluation of the torque of the incisor roots before and after the treatment.

Outcome (O): The influence of methods of incisor root torque control on the final effect of orthodontic treatment.

The following electronic databases were searched by two researchers (AK, JK): PubMed, EMBASE, and Cochrane Central Register of Controlled Trials [20], entering the following keywords:incisor retraction & orthodonticsincisor root torque control & orthodonticsroot torque & orthodonticsincisor inclination & orthodonticsincisor root torque control & retraction

Search filters include the time of publication of the article, the last 10 years, and publications that appeared in English in relation to conventional labial/buccal braces.

Based on the information provided in abstracts, articles were selected according to the following criteria: randomised clinical trials (RCTs) and controlled clinical prospective trials (CCTs). Individual case reports, case series reports, literature reviews, experimental studies, studies with limited data (including conference abstracts and journal writings), studies involving an unrepresentative group of patients (less than 10 patients), studies concerning patients with syndromes, and animal experiments were rejected. Articles unrelated to the subject of the planned study were also excluded.

In terms of the selected full-text articles, those that did not include information concerning the change in the inclination of the maxillary incisors after orthodontic treatment were excluded. Articles that did not report the number of patients who completed orthodontic treatment were also rejected.

For the remaining articles, references were reviewed, and such journals as *American Journal of Orthodontics*, *Dentofacial Orthopedics*, *International Orthodontics*, *Journal of Clinical Orthodontics*, and *Angle Orthodontist* were manually searched.

The following data were extracted from reviewed articles: year of publication, group size, characteristics of treatment and control groups, method of root torque control, maxillary incisor root torque before and after orthodontic treatment, together with mean changes in these angular values (Table 1).

## 3. Risk of Bias

The risk of bias analysis was performed for various articles using the Cochrane Collaboration’s tool [2]. The following criteria were used: random sequence generation, allocation concealment, blinding of participants and personnel, blinding of assessors, incomplete outcome data, reporting of selective outcomes and other potential sources of bias. A modified Newcastle-Ottawa Quality Assessment Scale [7] consisting of three parts was used for the qualitative assessment:(1)patient selection, where the following elements were evaluated:(a)the representativeness of the group exposed to the test agent,(b)the selection of patients for a control group,(c)the source of data concerning individual patients,(d)a demonstration that studied effects did not occur at the beginning of the study.

A maximum of 1 point was awarded for each sub-point, resulting in a possible score of 4 points.


(2)Confounding factors that evaluated whether a control group was identical to a treatment group in terms of other factors that could possibly influence the outcomes.


In this category, 0–2 points were awarded according to the significance of the influence of confounding factors.


(3)Outcome assessment, which analysed:(a)the blinding of assessors,(b)the duration of observation,(c)the percentage of patients who completed the study,


Enabling a maximum score of 3 points.

## 4. Statistical Analysis

For each article, a statistical analysis was performed for the differences in the mean changes in maxillary incisor inclinations between treatment and control groups. Studies with a statistically significant difference were selected. The outcomes are shown graphically as a forest plot (blobbogram). Moreover, a heterogeneity analysis of the included studies was conducted. For this purpose, a heterogeneity test based on the Q-statistic was performed, and I^2^ was calculated. All calculations were performed using Statistica 13 PLM software (StatSoft Poland, Krakow, Poland).

## 5. Results

By entering keywords into the included databases, 3175 abstracts were yielded. Forty-four articles were initially validated as eligible for the systematic review, and they were analysed in detail. A final total of 13 articles was selected, including 7 RCTs and 6 CCTs. The complete selection process is shown in Figure 1.

### 5.1. Group Size

The total number of participants was 580. The average group size was 20 patients. The largest groups were described by Chen et al. [2] and Xu et al. [8]: 32 patients per group. The smallest groups were reported by Deepak et al. [9]: 10 participants per group. In most studies, the sizes of the study and control groups were identical (Table 1).

### 5.2. Age and Sex

The ages of the patients varied significantly between articles, ranging from 10 years in [8] to 35 years in [7]. Therefore, some patients were treated before or during maximum growth, while others were treated upon reaching adulthood. The studies by Sadeki et al. [10], Lee et al. [11], Jiao et al. [12], and Jeea et al. [13] included only women. The studies conducted by Deepak et al. [9] and Ruan et al. [14] lacked information concerning patients’ sex. In each of the remaining studies, the female group was larger than the male group (Table 1).

### 5.3. Treatment Strategy

In all treated patients, extractions of the maxillary premolars were performed to gain space for the incisor and canine retraction. In 10 articles, maximum anchorage was used as TISAD in the treatment groups [8,9,10,11,14,15,16,17]. Other studies evaluated the effects of changes in the width of the slot inclination in brackets placed on the first upper molars [7], the use of corticotomy [17,19], and the use of elastics and power chains [12]. Those studies also compared the effectiveness of en-masse retraction to two-step anterior teeth retraction [8].

### 5.4. Risk Analysis

The outcome assessment findings for the risk of bias in the randomised clinical trials and the qualitative analysis of CCTs are shown in Table 2 and Table 3, respectively.

### 5.5. Main Parameter: Change in the Maxillary Incisor Root Torque

In all studies, incisor inclination was observed during the retraction movement, i.e., there was a vestibular root torque. In the treatment groups, the mean change in buccal-palatal inclination of the incisor roots was 10.46°. The greatest change in incisor torque (19.13°) was described by Lee et al. [11], in the group where mini-implants were used. Completely different results were obtained by Sadeka et al. [10] who found the smallest change, 4.41°, in the root inclination of the anterior teeth in a vestibulo-palatal direction after mini-implants were used on the vestibular side to retract the anterior teeth (Table 1).

Taking into account all studies, the mean difference in the upper incisor inclination between the control and treatment groups was 2.46°, which was statistically significant (*p* = 0.0003). The largest discrepancy between groups was observed by Davoody et al. [15]. In contrast, no discrepancy was observed between groups by Deepak et al. [9] (Table 1).

### 5.6. The Effectiveness of Methods for Upper Incisor Torque Control

Out of 13 articles [6,7,8,9,10,11,12,13,14,15,16,17,18,19] included in this review, the differences between study and control groups in upper anterior teeth inclination were statistically significant (*p* = 0.05) in only 6 articles. The analysis results are shown in a forest plot (blobbogram) [20] (Figure 2).

The results of the studies and their significant statistical value are shown below, in gradation from the study with the highest to the lowest statistical significance (Table 4).

(1)Al-Imam et al. [18] revealed that corticotomy during incisor retraction reduces their inclination by an average of 1.5° compared with non-surgically assisted retraction.(2)Al-Sibaie et al. [16] revealed that en-masse retraction of the anterior teeth using TISAD results in an incisor inclination on average 2.9° lower compared with the standard two-step process.(3)Sadeka et al. [10] revealed that the en-masse retraction using mini-implants and arches on the vestibular side results in a reduction of maxillary incisor inclination by an average of 5.85° compared with the same en-masse movement from the lingual access and mini-implants placed on the palate.(4)Chen et al. [2] revealed that patients treated with the PASS system had on average 4.8° lower maxillary incisor inclination compared with patients treated with the MBT system.(5)Zhao et al. [12] found that the use of intramaxillary elastics during incisor retraction results in an inclination of the maxillary anterior teeth that is on average 7.14° lower compared with the use of power chains.(6)Davoody et al. [15] found that the standard two-step retraction using an extra-intrusive arch results in a maxillary incisor inclination on average 8° lower compared with TISAD-assisted en-masse retraction.

Q was 36.25 with *p* = 0.0003 and I^2^ = 66.9%. Those data indicate a high level of study heterogeneity. This is most likely due to the different orthodontic techniques that were used in individual studies.

Additionally, an analysis of the heterogeneity of the studies included in the meta-analysis was performed. For this purpose, a test of heterogeneity based on Q and I^2^ was performed. The results are summarized in Table 5.

As is shown, I^2^ is 67%, which means that 67% of the differences observed between the test results are the result of heterogeneity. They are the result of differences between populations and the methods of torque control.

Table 6 shows the results of the heterogeneity of the studies broken down into the individual studies. The change in the cumulative effect shows how the result of meta- analysis would change if the study were not included. The percentage share of the study in the meta-analysis is the effect of the effectiveness of the torque control and the size of the study group (Deepak et al. studied only 10 people each in the study and control groups). The studies by Jeea, Davoody and Rua had the highest level of sensitivity and heterogeneity.

The last column includes the change in standard error, which shows how the combined effect would change, i.e., how a given study affects heterogeneity. The research by Al-Immam and Al-Sibaie had the greatest impact on heterogeneity in this aspect. 

The heterogeneity results are shown in Figure 3, Figure 4 and Figure 5.

## 6. Discussion

### 6.1. Risk of Bias

In most articles, the risk of bias was considered low due to the detailed, rigorous randomisation methods used. Only Tunçer et al. [17] described the risk of bias as moderate because the randomisation was not performed in a fully objective manner; instead, patients were assigned by the individual identification of their patient eligibility based on the criteria met.

The risk associated with the disclosure of group eligibility was considered low in all RCTs due to the use of opaque, sealed envelopes or other equivalent randomisation methods.

The use of specific treatments and the differences between them were known to both study participants and clinicians. Patients were aware of their participation in the study and signed an informed consent for proposed treatment. For this reason, the blinding of participants and personnel regarding treatment status was impossible, and the risk of bias for this criterion was identified as high.

Although the blinding of participants was not possible, the overall risk of bias was reduced in some studies by the blinding of assessors during the outcome analysis. In the articles by Al-Sibaie et al. [16], Al-Imam et al. [18], Chen et al. [2] and Tuncer et al. [17], assessors were not directly involved in the studies and did not know their purpose; therefore, the risk of bias was considered low. In the article by Sadeka et al. [10], assessors could easily define the purpose of the study (patients treated with vestibular or lingual orthodontic appliances), and thus blinding was not possible, and the risk was considered high. In contrast, Davoody et al. [15] provided no information on the blinding of assessors. However, they emphasised that assessors could easily identify patients in each group based on the analysis of cephalograms (presence or absence of mini-implants). A similar risk may have occurred in the study by Al-Sibaie et al. [16]. However, the authors clearly emphasised the blinding of assessors in their study. As it was difficult to determine the actual impact on risk of bias, it was finally considered uncertain in the article by Davoody et al. [15].

In all studies, complete data from patients were obtained and analysed. Therefore, the risk in this category was considered low.

In the articles by Davoody et al. [15] and Chen et al. [2], the authors highlighted the possibility of bias due to a reduction in the number of participants during the study. In the first study, the number of participants was reduced by approximately 30%, and in the second study by approximately 6%. Hence, the risk of selective reporting in the article by Davoody et al. [15] was identified as high, while in the article by Chen et al. [2] it was identified as moderate. In this category, the article by Tuncer et al. [17] also had a moderate risk. The reason was selectivity when selecting participants for the study.

Additional potential sources of bias were identified in several studies. In the article by Sadeka et al. [10], the authors highlighted that a different vertical position of mini-implants in the treatment groups, which was not included in the outcome analysis, may have influenced the inclination of the upper incisors. In contrast, Chen et al. [2] considered the influence of various anchorage methods that were used in individual patients as a potential source of bias in the results. Xu et al. [8] emphasised that the treatment of patients followed clinical standards, i.e., it was often tailored to the individual needs of patients and modified with the course of therapy. Therefore, a direct comparison of different treatments was not possible, which may have affected the lower statistical significance of the results obtained. Tuncer et al. [17] pointed out that the collection of molecular samples from patients started too late from the beginning of their study and that subsequent samples were taken at too-large intervals. In this case, however, these measurements were not relevant to this review. Finally, given the difficulty in determining the true impact of these limitations on the results obtained, the risk in the studies by Sadeka et al. [10], Chen et al. [2], and Xu et al. [8] was considered unclear.

The articles by Deepak et al. [9], Lee et al. [11] and Koyama et al. [18] received 4 points (maximum) for the patient selection criterion. Other articles (Ruan et al. [14], Jeea et al. [13]) lost 1 point due to lack of control. Because of confounding factors, also due to lack of control, the articles by Ruan et al. [14] and Jeea et al. [13] did not receive points, while other articles received the maximum number of points (2 points). Regarding the last criterion (evaluation of study effects), all articles lost 1 point due to the lack of blinding of assessors. Moreover, the study by Ruan et al. [14] lost 1 point due to a significant reduction in the number of patients eligible for the outcome analysis compared with the initial group. Finally, the study in question received 1 point in this category.

In addition to the discussed methods of controlling the axial inclination of incisors at the time of closing the post-extraction hatches, non-orthodontic methods are also mentioned such as corticotomy. According to the researchers, the use of these methods can also affect the rate as well as the range of the tilting of the incisors [21]. Adequate influence on the metabolism of specific areas of the compact bone can affect the distribution of the bone support of the teeth and thus the location of both the centre of resistance and the centre of rotation of the teeth. This may result in easier control of the torque of the front teeth during their retraction. Similarly, Zedeh et al. [21] showed, access to the subperiosteal vestibular incision tunnel (VISTA) for surgically assisted orthodontic treatment (SFOT) may improve the control of the axial position of the incisors during extraction therapy [21].

### 6.2. Outcome Analysis

There are many different methods for controlling root inclination during the retraction of the anterior teeth; however, most of them have not been sufficiently analysed, and hence, their effectiveness is not fully validated. According to the current systematic review, not all methods of incisor inclination control differ in terms of performance in a statistically significant way.

In terms of the methods of torque control that were reported in this systematic review, those with the highest statistically validated efficacy are worth analysing.

The use of corticotomy when retracting anterior teeth significantly reduces maxillary incisor inclination [19], and corticotomy may be relevant to root inclination control, i.e., incisions should be made on both the vestibular and palatal sides. Al Ihmam [19] revealed that torque control after corticotomy was good; that study had high-value evidence. In Tunçer et al.’s study. [17], in which incisions were made only on the vestibular side, there was no statistically significant difference in the loss of control of incisor root inclination compared with a control group. Therefore, the study revealed that torque control was not good enough when limiting corticotomy to the vestibular side only; that study’s evidence had moderate value. Given the above-mentioned analyses, it can be concluded that the possibility of torque control using corticotomy was proven; however, it requires additional studies on a larger group divided into control and treatment groups, and it requires a comparison between corticotomy performed only on the vestibular side and that performed on both the vestibular and palatal sides. 

The use of TISADs during retraction also significantly reduces the maxillary incisor inclination (loss of control over the buccal–palatal inclination of the incisor roots). This is because the vector of force used for retraction approaches the centre of resistance of the teeth more closely than, e.g., during the standard retraction, where the force is applied to brackets placed on the maxillary molars. Furthermore, the standard retraction is often performed in two steps, which may further affect the greater retroclination of the incisors. This is confirmed by Al-Sibaie et al. [16] The value of the evidence for these studies is high. In this respect, it can be concluded that the torque control using TISAD is proven; however, as only one study was conducted, additional comparative studies will be necessary to unanimously accept this thesis. 

When retracting the anterior teeth, it is more advantageous to use vestibular than lingual mechanics [10]. The en-masse retraction using mini-implants and arches on the vestibular side results in a reduction of maxillary incisor inclination compared with the same en-masse movement from the lingual access and mini-implants placed on the palate. In the latter, the force vector lies farther from the centre of resistance of the retracted segment, which causes the teeth to be inclined more strongly and torque control to become weaker. Those issues were addressed by Sadek et al. [10], whose study evidence had moderate value. A shortcoming of that study is that the authors did not consider the outcome analysis of the different vertical positions of the mini-implants in the analysed groups, which may affect the inclination of the upper incisors and make a precise comparison of torque controls impossible. With these concerns in mind, detailed studies taking into account the reproducibility of mini-implant placement should be conducted to obtain reliable findings that describe the effects of this type of anchorage and its placement on the final inclination of the incisor long axis. When considering the studies concerning the use of mini-implants to control incisor inclination during retraction by acknowledging some imperfections in the studies conducted but taking into account their high-value evidence, this method can be considered proven.

The use of management protocols such as the intrusive arch [10,15] or the PASS [7] system during incisor retraction should be mentioned. Brackets placed on the molars in the PASS system have an extra slot with an inclination angle of 25°. The insertion of an extra arch into this slot and its attachment to the incisors have a similar effect to the use of an intrusive arch: It strengthens the anchorage in the molar region and causes the intrusion of the incisors and their protrusion, thus increasing the root inclination control during retraction. The study used an intrusive arch both combined with mini-implants [10] and without mini-implants [15], with better results for torque control when a skeletal anchorage was included. Nevertheless, the use of the intrusive arch alone in anterior teeth retraction [15] turned out to be more effective in controlling root inclination compared with isolated skeletal anchorage [16]. This emphasises the role of the control of the vertical dimension during retraction in maintaining the correct incisor root inclination. The evidence from the studies in question was of moderate value. Due to the small sizes of the treatment groups, it should be concluded that the above-mentioned methods are effective in terms of incisor torque control; however, further studies of larger groups are necessary to unequivocally prove the superiority of these methods over others. 

The use of class I elastics results in less incisor inclination during retraction compared with the use of a power chain for this purpose [12]. This difference is probably due to the significantly lower force acting on the anterior segment when using intramaxillary elastics (approx. 100 g) instead of a power chain (approx. 250 g).

The effect of age on the degree of incisor inclination control is probably minor, as suggested by Ruan et al. [14]. Although the retroclination of the incisors in that study was greater in the adult group, there was no statistically significant difference between groups at different ages. The value of evidence for that study is considered low, and thus the age criterion cannot be considered a factor that influences the change in incisor inclination during orthodontic treatment involving premolar extraction. 

The statistical analysis revealed that the use of additional components that control root inclination resulted in less incisor inclination during retraction compared with closing gaps after missing teeth without the use of these methods (Table 4). Nevertheless, not all methods were equally effective. The greatest difference between the control group and the treatment group assisted by control mechanisms of root inclination was found for the standard two-step retraction using the extra-intrusive arch (Table 4). In terms of comparing the effectiveness of torque control during en-masse retraction in various studies and ranking it from highest to lowest, the following order of studies was obtained: Sadeka et al. [10], Li et al. [12], Chen et al. [2], Al-Sibaie et al. [16], and Al-Imam et al. [18] (Table 1). It should be noted that the scientific credibility of those studies, ranked from largest to smallest, had a different order: Al-Imam et al. [18], Al-Sibaie et al. [16], Chen et al. [2], Sadeka et al. [10], Davoody et al. [15] and Zhao et al. [12] (Table 2 and Table 3).

The effect on the combined result of the statistical analysis also varied. In descending order from the highest aggregate score to the lowest, the ranking of studies was as follows: Al.-Imam et al. [18], Al-Sibaie et al. [16], Sadeka et al. [10], Chen et al. [2], Zhao et al. [12] and Davoody et al. [15] (Table 4). In conclusion, the results of studies that prove the high effectiveness of root inclination control are, unfortunately, often associated with low statistical reliability. However, it can be assumed that the use of corticotomy involving incisions made on both the vestibular and palatal sides, the en-masse retraction of the anterior teeth using TISAD placed on the vestibular side, and the use of the extra-intrusive arch provide the best incisor torque control. This does not preclude the need for more detailed studies of larger control groups to demonstrate greater differences in terms of methods used. 

## 7. Limitations

The main limitations of this review include the time-period restriction, the last 10 years, and the restriction to English-language publications. This may have had an impact on the risk of statistical bias in the publication. Another limitation was the low number of clinical trials (RCTs and CCTs) that analysed the topic addressed. The inclusion of only articles that describe studies concerning the torque value both before and after orthodontic treatments proposed according to specific treatment protocols also became a limitation.

In terms of studying the effectiveness of different methods of teeth retraction, the presence of a control group proved to be very important. By definition, a control group should not receive treatment. Therefore, it is not possible to establish a control group when examining the change in root inclination of the anterior teeth because among other reasons, orthodontic treatment cannot be dispensed with after an extraction for orthodontic indications. Consequently, this study is a comparison of different methods of torque control. In the article selection process, some clinical trials were considered equivalent to control studies because the results showed a comparison of treatment groups despite differences in the article description. The review also included one article that was a retrospective study but met the requirements for CCTs. The statistical analysis revealed that a significant SD in the absence of large study and control groups casts doubt in many articles on the reliability of results obtained. This explains the need for further studies, especially RCTs with more homogeneous groups of sufficiently large size.

## 8. Conclusions

The analysed studies indicate that corticotomy or orthodontic mini-implants are the most effective and scientifically proven methods of incisor inclination control during incisor en-masse retraction [16,19]. The use of both vestibular mechanics and an extra slot in brackets placed on the molars, which creates an effect that is similar to that of the intrusive arch, was studied in protocols with unclear risk of bias where various factors could affect the reliability of the results [7,10,15]. In contrast, the use of light force of intramaxillary elastics was only analysed in a non-randomised clinical trial. Therefore, the conclusions of the present study should be interpreted with caution (Table 3). The patient’s age also seems to be irrelevant to torque control [14].

Although all studies reported that the incisors were inclined after retraction, it should be noted that extraction is often performed when the incisors are excessively inclined. Hence, treatment aims to partially straighten the incisors and slight incisor inclination cannot be considered as a lack of control of the buccal-palatal inclination of incisor roots provided that it is not excessive. To give a definitive answer to the question concerning which method most effectively controls torque during anterior teeth retraction, further high-quality RCTs studies.

## Figures and Tables

**Figure 1 diagnostics-12-01611-f001:**
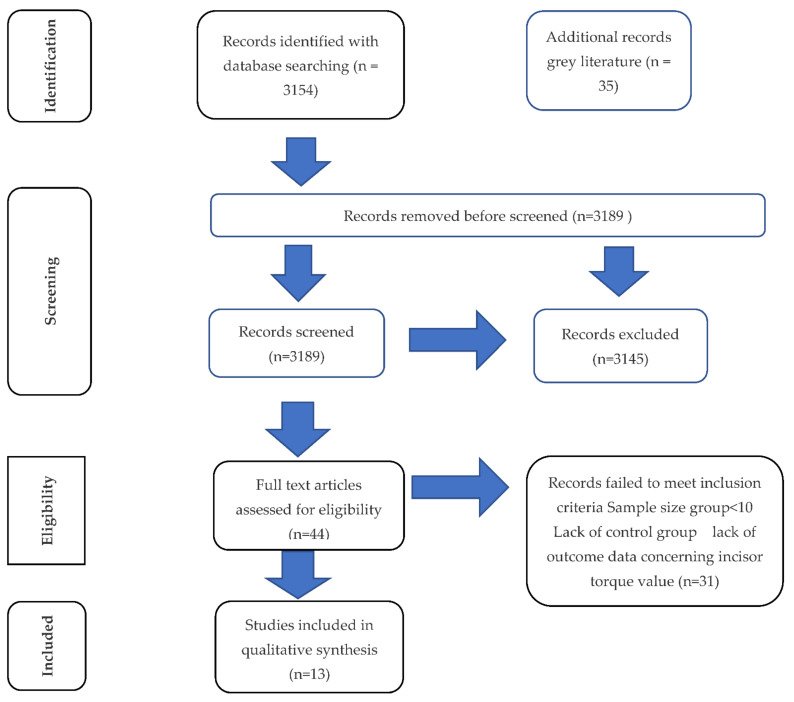
PRISMA flow diagram.

**Figure 2 diagnostics-12-01611-f002:**
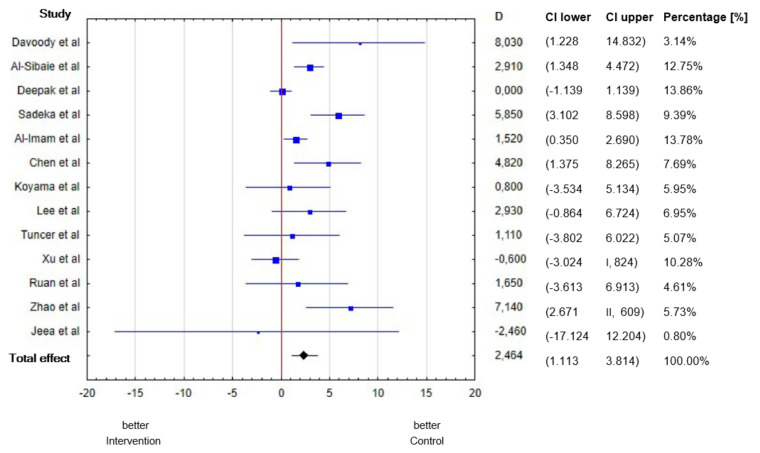
The mean differences in the incisor inclination between the treatment and control groups.

**Figure 3 diagnostics-12-01611-f003:**
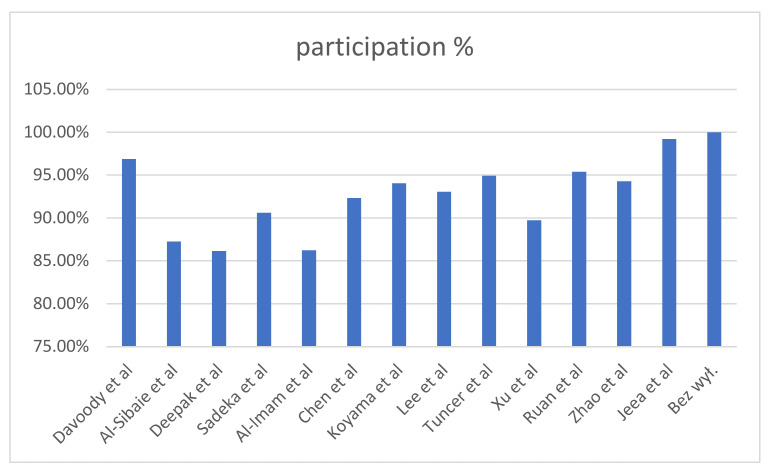
Heterogeneity analysis results by study.

**Figure 4 diagnostics-12-01611-f004:**
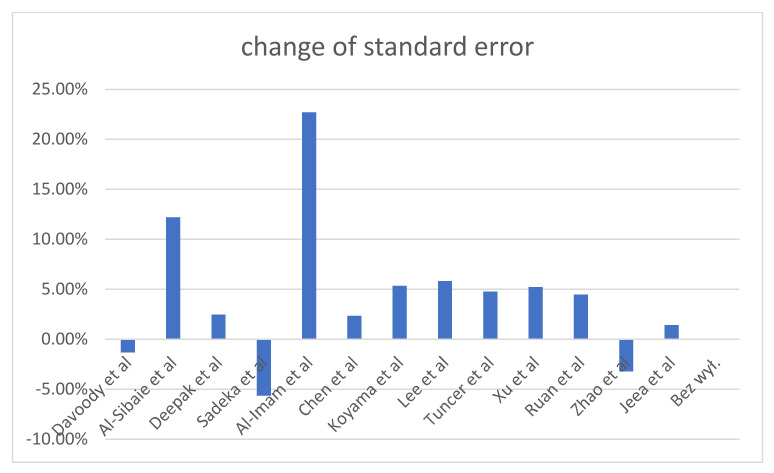
Heterogeneity analysis: study impacts on heterogeneity.

**Figure 5 diagnostics-12-01611-f005:**
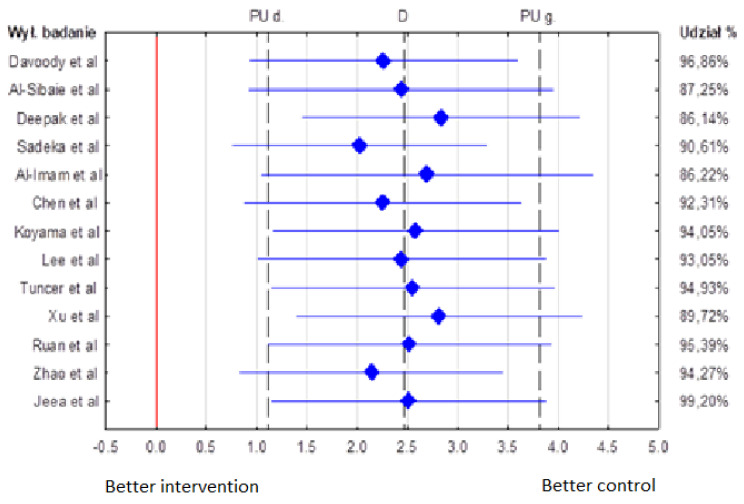
Heterogeneity analysis results.

**Table 1 diagnostics-12-01611-t001:** Data from articles.

Study	Control Group	Study Group	Age of Participants (yrs)	Treatment Strategy	Change in the U1-Inclination during the Observation Period (°)	SD (°)	Median Difference of U1-Inclination between Groups (G1–G2, °)	SE (Standard Error)	*p*	Contribution for Analysis Result (%)
RCTs										
Davoody et al. (2012) [15]	G1: 15 (M5 F10)	G2: 13 (M7 F6)	Median 17–18	G1: 2-step retraction; G2: retraction with TADs	U1-X; G1: −10.63; G2: −18.66	G1: 9.9 G2: 8.21	8.02	3.471	0.0207	3.14
Al-Sibaie and Hajeer (2013) [16]	G2: 28 (M12 F16)	G1: 28 (M9 F19)	Median 20–23	G1: retraction with TADs; G2: 2-step retraction with TPA	U1-Sn; G1: −5.03; G2: −7.94	G1: 3.39 G2: 2.51	2.91	0.797	0.0003	12.75
Sadeka et al. (2019) [10]	G1: 14F	G2: 14F	Median 20 ± 2	G1: retraction with TADs, buccal mechanics; G2: retraction with TADs, lingual mechanics	U1-PP; G1: −4.41; G2: −10.26	G1 2.33 G2: 4.7	5.85	1.402	0	9.39
Al-Imam et al. (2019) [18]	G2: 20 (M4 F16)	G1: 20 (M5 F15)	Median 19.5 (16–31)	G1: 2-step retraction with corticotomy; G2: 2-step retraction without corticotomy	U1-Sn; G1: −7.88; G2: −9.40	G1: 2.28 G2: 1.39	1.52	0.597	0.0109	13.78
Chen et al. (2020) [2]	G2: 32 (M11 F21)	G1: 32 (M10 F22)	11–35	G1: retraction with PASS; G2: retraction with MBT	G1: −6.94; G2: −11.76	G1: 6.35 G2: 7.65	4.82	1.758	0.0061	7.69
Tunçer et al. (2017) [17]	G2: 15 (M2 F13)	G1: 15 (M2 F13)	14<	G1: retraction with TADs and piezosurgery; G2: retraction with TADs without piezosurgery	U1/HRP; G1: −8.87; G2: −9.98	G1: 5.38 G2: 8.08	1.11	2.506	0.6579	5.07
Xu et al. (2010) [8]	G1: 32 (M12 F20)	G2: 31 (M12 F19)	10–16	G1: retraction with TADs; G2: 2-step retraction	U1 ling crown tipp.; G1: −10.7; G2: −10.1	G1: 5.1 G2: 4.7	−0.6	1.237	0.6276	10.28
CCTs										
Deepak et al. (2014) [9]	Gk: 10 (sex NR)	G1: 10 (sex NR)	14–25	G1: retraction with TADs; Gk: retraction with posted wires and springs from first molars	G1: −5.8; Gk: −5.8	G1: 1.3 Gk: 1.3	0	0.581	1	13.86
Koyama et al. (2011) [19]	G2: 14 (M2 F12)	G1: 14 (M1 F13)	Median 24.9 ± 5	G1: retraction with TADs; G2: retraction with a headgear appliance	SN-U1; G1: −10.3; G2: −11.1	G1: 5.8 G2: 5.9	0.8	2.211	0.7175	5.95
Lee and Kim (2011) [11]	G1: 20F	G2: 20F	Median 23.32	G1: retraction with a headgear appliance; G2: retraction with TADs	U1-PP; G1: −16.20; G2: −19.13	G1: 5.59 G2: 6.61	2.93	1.936	0.1301	6.95
CTs										
Ruan et al. (2018) [14]	G1: 10 (sex NR)	G1: 19 (sex NR)	G1: adolescents; G2: adults	G1, G2: retraction with TADs	Tor1; G1: −9.82; G2: −11.47	G1: 8.97 G2: 6.70	1.65	2.685	0.5389	4.61
Zhao et al. (2018) [12]	G1: 18F	G2: 21F	15<	G1: retraction with elastics (TADs for anchorage control); G2: retraction with power chains (TADs for anchorage control)	U1-Sn; G1: −8.84; G2: −15.98	G1 8.53 G2: 5.60	7.14	2.28	0.0017	5.73
Jeea et al. (2013) [13]	G1: 15F	G2: 16F	Median 21	G1: retraction with a conventional C-wire and TADs; G2: retraction with a preformed C-wire and TADs	U1-Sn; G1: −13.77; G2: −11.31	NR	−2.46	7.482	0.7423	0.80

**Table 2 diagnostics-12-01611-t002:** Risk of bias assessment in the RCTs.

Study	Random Sequence Generation	Allocation Concealment	Blinding of Participants and Personnel	Blinding of an Outcome Assessment	Incomplete Outcome Data	Selective Reporting	Other Bias
Al-Sibaie and Hajeer (2013) [16]	Low	Low	High	Low	Low	Low	Low
Davoody et al. (2012) [15]	Low	Low	High	Unclear	Low	High	Low
Sadeka et al. (2019) [10]	Low	Low	High	High	Low	Low	Unclear
Al-Imam et al. (2019) [18]	Low	Low	High	Low	Low	Low	Low
Chen et al. (2020) [2]	Low	Low	High	Low	Low	Moderate	Unclear
Tunçer et al. (2017) [17]	Moderate	Low	High	Low	Low	Moderate	Low
Xu et al. (2010) [8]	Low	Low	High	Low	Low	Low	Unclear

**Table 3 diagnostics-12-01611-t003:** The quality assessment of CCTs according to the modified Newcastle–Ottawa Scale.

Study	Selection	Comparability	Outcome Assessment
Deepak et al. (2014) [9]	4	2	2
Lee and Kim (2011) [11]	4	2	2
Koyama et al. (2011) [19]	4	2	2
Ruan et al. (2018) [14]	3	-	1
Zhao et al. (2018) [12]	3	-	2
Jeea et al. (2013) [13]	3	-	2

**Table 4 diagnostics-12-01611-t004:** The most efficient treatment strategies of anterior torque control during retraction.

Study	Treatment Strategy	Median Difference of U1-Inclination between Groups (G1–G2, °)	*p*	Contribution for Analysis Result (%)
Al-Imam et al. (2019) [18]	G1: 2-step retraction with corticotomy; G2: 2-step retraction without corticotomy	1.52	0.0109	13.78
Al-Sibaie and Hajeer (2013) [16]	G1: retraction with TADs; G2: 2-step retraction with TPA	2.91	0.0003	12.75
Sadeka et al. (2019) [10]	G1: retraction with TADs, buccal mechanics; G2: retraction with TADs, lingual mechanics	5.85	0	9.39
Chen et al. (2020) [2]	G1: retraction with PASS; G2: retraction with MBT	4.82	0.0061	7.69
Zhao et al. (2018) [12]	G1: retraction with elastics (TADs for anchorage control); G2: retraction with power chains (TADs for anchorage control)	7.14	0.0017	5.73
Davoody et al. (2012) [15]	G1: 2-step retraction; G2: retraction with TADs	8.02	0.0207	3.14

**Table 5 diagnostics-12-01611-t005:** Heterogeneity analysis.

Q	df	*p*	I^2^	Lower Limit 95% PU (I^2^)	Upper Limit 95% PU (I^2^)
36,249	12	0.0003	66.90%	40.72%	81.51%

**Table 6 diagnostics-12-01611-t006:** Heterogeneity analysis results; changes in the cumulative effects.

Study	D	Standard Error	Lower Limit 95% PU	Upper Limit 95% PU	*p*	Participationł %	Change of Standard Error
Davoody et al. [15]	2.2687	0.6798	0.9363	3.6011	0.001	96.86%	−1.33%
Al-Sibaie et al. [16]	2.4392	0.7731	0.9240	3.9545	0.002	87.25%	12.21%
Deepak et al. [9]	2.8349	0.7060	1.4511	4.2186	0.000	86.14%	2.47%
Sadeka et al. [10]	2.0259	0.6500	0.7519	3.2999	0.002	90.61%	−5.66%
Al-Imam et al. [18]	2.6975	0.8454	1.0407	4.3544	0.001	86.22%	22.69%
Chen et al. [2]	2.2559	0.7052	0.8738	3.6380	0.001	92.31%	2.35%
Koyama et al. [19]	2.5864	0.7259	1.1637	4.0090	0.000	94.05%	5.35%
Lee et al. [11]	2.4437	0.7291	1.0146	3.8728	0.001	93.05%	5.83%
Tuncer et al. [17]	2.5522	0.7218	1.1374	3.9669	0.000	94.93%	4.76%
Xu et al. [8]	2.8121	0.7250	1.3912	4.2331	0.000	89.72%	5.22%
Ruan et al. [14]	2.5183	0.7198	1.1075	3.9290	0.000	95.39%	4.47%
Zhao et al. [12]	2.1462	0.6666	0.8396	3.4529	0.001	94.27%	−3.24%
Jeea et al. [13]	2.5113	0.6987	1.1419	3.8807	0.000	99.20%	1.41%
Bez wył.	2.4636	0.6890	1.1132	3.8140	0.000	100.00%	0.00%

D—difference in means.

## Data Availability

The data sets that are used and/or analysed during the current study are available from the corresponding author upon reasonable request.

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
