# Peer review of "Methods of Anterior Torque Control during Retraction: A Systematic Review"

_diagnostics, 2022, doi:10.3390/diagnostics12071611_

Round 1

Reviewer 1 Report

Please provide bibliography for this paragraphs -There is a need for the mechanical control of incisor root position in the treatment of 
moderate to severe crowding, bimaxillary protrusion, open bite or class II malocclusion. 
Extraction of the premolars to retract the canines is often required when planning orthodontic treatment for defects in question. Spaces occurring mesially to distalised canines 
make the incisors particularly susceptible to uncontrolled/excessive inclination. To avoid 
this effect, such different methods of incisor torque control were suggested as brackets 
with increased built-in torque, arch torsion, placement of temporary intraoral skeletal anchorage devices (TISAD) that enable group distal movement of the "social six" (en-masse 
retraction), or a mini implant inserted between the maxillary incisors. Regardless of various procedures that support proper root position during space closure, the evaluation of 
their effectiveness is based almost exclusively on individual clinicians' experience.-

Please perform a new prisma because it is not complete.

The conclusion is too long. It should be one paragraph and more concise.

The authors could take informations also from this article 

DOI

10.3390/app12052657

Author Response

Dear Reviewer. Thank you very much for your comments and attention.

  1. The translation of the text took place by a licensed translation agency and native speacker
  2. Bibliography attached to proposed paragraphs
  3. A new PRISMA was made
  4. Conclussions have been shortened and modified
  5. The proposed article has been taken into account

Best Regards.

Reviewer 2 Report

The topic is very interesting but the study has decisive methodological errors

Introduction

1st paragraph the reference is missing

“…There is a need for the mechanical control of incisor root position in the treatment of moderate to severe crowding, bimaxillary protrusion, open bite or class II malocclusion…” - the reference is missing

The penultimate paragraph has no reference.

How many authors did the scrutiny of the articles.

Materials and Methods

The research methodology should be redone, since it has numerous flaws, jeopardizing the quality of the systematic review. The search must be in natural and controlled language. Authors must include all search keys used for the various databases. The prism flow diagram is incomplete and out of date. Authors should search the gray literature.

Why did the authors not consider doing research on the Web of Sciences?

PICO is not a study design, but a formulated question that should be answered in the conclusions of the systematic review. Authors should correct this flaw.

Placing a time limit is not an inclusion criterion. It is part of the search and is called a search filter. Authors must correct the error.

As well as the language of the article. This is a filter and not an exclusion criterion.

Results

Table 2 and 3 should be in graphical format for better understanding.

Conclusions

Conclusions should be shorter, should result from the authors' analysis of the results after having conducted the discussion, and should not repeat the discussion.

Author Response

Response for Reviewer 2

Dear Reviewer. Thank you very much for your comments and attention.

  1. The translation of the text took place by a licensed translation agency and native speaker
  2. Bibliography attached to proposed paragraphs
  3. The number of researchers reviewing the literature is taken into account
  4. The search of the database concerned literature in English
  5. The search uses the same search keys and filters
  6. The authors searched the available databases. Due to the limited time period between obtaining a review and responding to a review, limited access to grey literature, phDs and unpublished material has been taken into account.
  7. PICO - the suggestion was taken into account and the questions answered in the conclusions were changed
  8. It seems that the results in Tables 2 and 3 are clear and we would stick to this concept.
  9. The request was granted. Conclusions have been shortened and modified

Best Regards

Reviewer 3 Report

Interesting paper

Very deficient introduction, you need to expand this section and very briefly add methods of anterior torque control during retraction such as TADs (read cite,J Clin Orthod. 1997 Nov;31(11):763-7.) 

Tables, add the study sample and numbers for control and the other group

Discussion, you need to talk about the advances in using TADs in orthodontics, new methods, in particular minimally invasive approaches (Japanese Dental Science Review 2022, 58;137-154; Contemp Clin Dent. 2019 ;10(3):548-553.)

In conclusion, very long shorten, revise, what is the recommended method, remove references in this section, and only talk about significant findings and recommendations for future research

Author Response

Dear Reviewer. Thank you very much for your comments and attention.

  1. The translation of the text took place by a licensed translation agency and native speaker
  2. The topic of our analysis was not to discuss the use of non-orthodontic/surgical methods supporting tooth shifting. Therefore, we could not include these articles directly in the analysis. Nevertheless - because this issue is very interesting and important, we have discussed it using the articles you indicated, both in the introduction and in the discussion.
  3. The authors did not include the proposed articles in the analysis, because they are not orthodontic methods, but surgical methods supporting orthodontic treatment and can also be combined with other methods. In the opinion of the authors, they deserve a separate analysis of their effectiveness as methods supporting torque control in extraction treatment.
  4. The request was granted. Conclusions have been shortened and modified

Best Regards

Round 2

Reviewer 2 Report

The authors made the methodology changes that I recommended, and that if they were not carried out, their recommendation for publication would be unfeasible. Once they have all been carried out, only one issue for amendment can be considered. The elaboration of the PICO question. The authors have in each of the items of the PICO question a question. This is not what is intended, but in each item the identification of the defendant (eg population: adult patients using fixed apparatus). At the end of this identification, the question must be constructed and peaked based on the information gathered from each item.

Author Response

Dear Reviewer Thank you for your attention and time. In the text of the manuscript we tried to include the suggested changes.

Best Regards

Reviewer 3 Report

the paper has improved but needs more revision

introduction, second page, 2nd paragraph, 'use of orthodontic microimplants' correct to'use of orthodontic mini-implants ot Temporary Anchorage Devices (TADs)'

next paragraph, 2nd line, 'class II' correct to'Class II"

same paragraph, next line'Extraction of the premolars to retract the canines is often required when planning orthodontic treatment for defects in question', revise does not make sense. again line 9''a mini implant' correct to'a mini-implant'

objectives, define if this systemic review aimed at conventional labial /buccal braces, aligners, or lingual braces, as lingual systems are very susceptible to retroclination during retraction

results, table 1, can you add the 95% CI of the difference in incisor inclination , instead of standard error and the last 3 columns, what bracket system did they use, 018 or 022

discussion, talk about the study sample sizes, how many were 30 or above in each group, as this is the ideal sample in each group to create a  good 95% CI

references, references 21 and 23 are the same

Author Response

Dear Reviewer Thank you for your attention and time.

In the text of the manuscript we tried to include the suggested changes.

  1. The word microimplant has been changed to "use of orthodontic miniimplants or temporary anchor devices (TAD)"
  2. The word Class II malocclusion has been changed to class II malocclusion
  3. The wording "Extraction of premolars to retract canines is often required when planning orthodontic treatment of given defects" has been corrected and revised to receive a new wording
  4. In line 9, the word miroimplant was replaced by a mini – implant
  5. In the discussion of the objectives, it was specified that the current systematic review applies to conventional labial/cheek braces.
  6. Dear Reviewer. Thank you for the valuable comments you have submitted to clarify the results of the research. However, the work we have presented is a review of the available literature and only discusses the results and methodology included in the published articles. Therefore, we cannot comply with your request in this regard.
  7. Dear Reviewer. We are aware that the correct group size to determine the 95% confidence level is 30 people. Unfortunately, the data in the literature in question do not precisely determine the size of the group.
  8. References 21 and 23 have been massed and unified in the text.

Best Regards